# Analysis of sensitive information leakage in functional genomics signal profiles through genomic deletions

Arif Harmanci [1,2,3] & Mark Gerstein [1,2,4]

Functional genomics experiments, such as RNA-seq, provide non-individual specific information about gene expression under different conditions such as disease and normal. There is great desire to share these data. However, privacy concerns often preclude sharing of the raw reads. To enable safe sharing, aggregated summaries such as read-depth signal profiles and levels of gene expression are used. Projects such as GTEx and ENCODE share these because they ostensibly do not leak much identifying information. Here, we attempt to quantify the validity of this statement, measuring the leakage of genomic deletions from signal profiles. We present information theoretic measures for the degree to which one can genotype these deletions. We then develop practical genotyping approaches and demonstrate how to use these to identify an individual within a large cohort in the context of linking attacks. Finally, we present an anonymization method removing much of the leakage from signal profiles.

[1] Program in Computational Biology and Bioinformatics, Yale University, New Haven, CT 06520, USA. [2] Department of Molecular Biophysics and Biochemistry, Yale University, New Haven, CT 06520, USA. [3] School of Biomedical Informatics, Center for Precision Health, University of Texas Health Science Center, Houston, TX 77030, USA. [4] Department of Computer Science, Yale University, New Haven, CT 06520, USA. Correspondence and requests for materials should be addressed to A.H. (email: arif.o.harmanci@uth.tmc.edu) or to M.G. (email: mark@gersteinlab.org)

ndividual privacy is emerging as an important aspect of biomedical data science[1]. Large research projects will create deluge of genetic data from millions of individuals[2–6]. The leakage of genetic information may create privacy concerns such as discrimination by health insurance companies[7]. Initial studies on genomic privacy focused primarily on protecting the identities of participants in genetic studies[8,9]. These focused on how an individual's genetic information can be used to reliably predict whether they participated in a particular genetic study. In this arena, differential privacy[10] has been proposed as a theoretically optimal formalism such that the probability that any individual's participation can be identified can be made arbitrarily small. In addition, cryptographic approaches have proven useful for privacy-aware analyses of genomic datasets, albeit with high requirements of computational resources[11–13].

The declining cost of DNA sequencing has led to increase in the number of available genomic datasets. This increase will render linking attacks[14–16] more practical where an adversary can use statistical methods to link multiple datasets to reveal sensitive information. Briefly, linking attacks are based on crossreferencing of two or more datasets that are released independently. Some of the datasets contain personal identifying information (e.g., names or addresses), while others contain sensitive information (e.g., health information). The immediate consequence of cross-referencing is that sensitive information from one dataset becomes linked to the identifying information in another and this, in turn, breaches the privacy. The risks behind linking attacks have risen recently because large volumes of personal data are independently released and maintained. Consequently, risk assessment is complicated because a dataset that is currently deemed safe to release may become a target for linking attacks when another dataset is released in the future. A well-known example of a linking attack is the Netflix Prize Competition[14] (Supplementary Fig. 1a, b, Supplementary Note 1) where many participants were affected by a linking attack. Similar attacks will be major routes to genomic privacy breaches.

In this study, we analyze the leakage of sensitive information from functional genomics data in linking attacks. Functional genomics data, such as those from RNA sequencing (RNA-Seq), is unique in the following sense: the raw reads contain genetic information and they could be used to identify individuals (Supplementary Fig. 2b). However, the main purpose of RNA-Seq data is not related to the variants, but rather understanding how the activity of genes changes under different conditions such as cancer. Thus, unlike the variant data, functional genomics datasets have more complicated "Yin-Yang" aspect with relation to privacy. In addition, functional genomics datasets are sometimes shared with phenotypic information that is potentially of private value (e.g., a particular condition or disease that a person has). This leads to an interesting situation where the data is ostensibly collected and used for non-personal purposes to determine general aspects about a condition. However, the existence of small amounts of private information in the data can be revealing about the individual from which they came.

On the other hand, there is great desire to share these data openly to understand diseases such as cancer. Although the raw reads cannot be shared, there is a general belief that aggregated data computed using raw reads, such as signal profiles and genelevel quantifications, can be shared. In fact, several large consortia, for example the ENCODE project[17], the Roadmap Epigenome Mapping Consortium[18], and GTEx[19,20], share personal RNA-seq signal profiles publicly while the genotypes are under restricted access (Supplementary Fig. 3). Signal profiles reflect the overall depth of read coverage at any given position on the genome (Supplementary Fig. 2a). In this study, we focus on leakage from signal profiles. Another commonly shared aggregated data are gene-level quantifications, which are essentially averages of the signal profile over exons. The leakage from these has been explored elsewhere[15,21].

Several studies examined aspects of linking attacks in genomic privacy[15,16,21,22]. These studies focus on single nucleotide variants (SNVs) because the estimated regulatory effect of SNVs on gene expression is much larger than structural variants (SVs)[23]. However, the major portion of genomic variation, in terms of the number of affected nucleotides, is caused by SVs[24,25], as shown by The 1000 Genomes Project. We expect the phenotypic change caused by an SV to be much larger than an SNV. For example, homozygous deletion of a gene will cause the total disappearance of its expression. Here, we explore whether an adversary could use functional genomics signal profiles (such as RNA-seq) to detect and genotype genomic deletions and use them to pinpoint individuals in a linking attack. Signal profiles are currently at the junction of open and restricted data sharing where genomic data has begun to be shared publicly. Hence, it is particularly important to probe the leakage from the signal profile representation of functional genomics data. We emphasize that we are not trying to look at all sources of leakage from functional genomics data, but just the sources right at the boundary of shareable and nonshareable data.

In this paper, we show an adversary can detect small and large genomic deletions in signal profiles and we present metrics for predictability of deletion genotypes. We highlight two quantities that determine how well genomic deletions can be detected from sequencing data. These are breadth of coverage and depth of coverage, respectively. We analyze RNA-seq[26] and ChIP-Seq[27] profiles for information leakage. RNA-seq is concentrated on exonic regions and has high depth of coverage but low breadth of coverage. We show that they can be used for genotyping small deletions. On the other hand, ChIP-Seq signal profiles generally have high breadth of coverage but low depth of coverage and can be used for detecting large deletions. We show that these signal profiles can be used for practical and successful linking attacks and we also present an anonymization method for protecting RNA-seq signal profiles.

## Results

**Linking attack scenario**. Figure 1 summarizes the linking attack scenario (Supplementary Note 2). The attack involves crossreferencing the individuals in a signal profile dataset, $S$, against the individuals in a genotype dataset, $G$. The signal profile dataset is publicly available for research purposes and contains a genome-wide signal profile and an anonymized identifier for each individual. The signal profile for an individual represents the measurements of activity at each genomic position. This dataset stores a genome-wide signal profile for each individual, for example containing RNA-Seq or ChIP-Seq data. In addition, the signal profile dataset contains sensitive information about each individual (e.g., HIV status). The genotype dataset, $G$, contains the genotypes of a panel of SV, denoted by $P_G$. $G$ also contains the identities of the individuals and it is restricted access. The adversary obtains access to $G$ by lawful or unlawful means, e.g., adversary might have stolen it or might be allowed to access it but violated the terms of accession (e.g., "variants from a glass"). The main objective of the adversary is to link $G$ and $S$ by first predicting the genotypes using signal profiles in $S$, and then matching the predicted genotypes to the genotypes in $G$. For any matching individuals in $G$ and $S$, the name and sensitive information are revealed to the adversary.

The attack has two steps. The first step is genotyping the deletion variants, which is illustrated in Fig. 1a. In the first scenario, we assume that the adversary has access to a reference

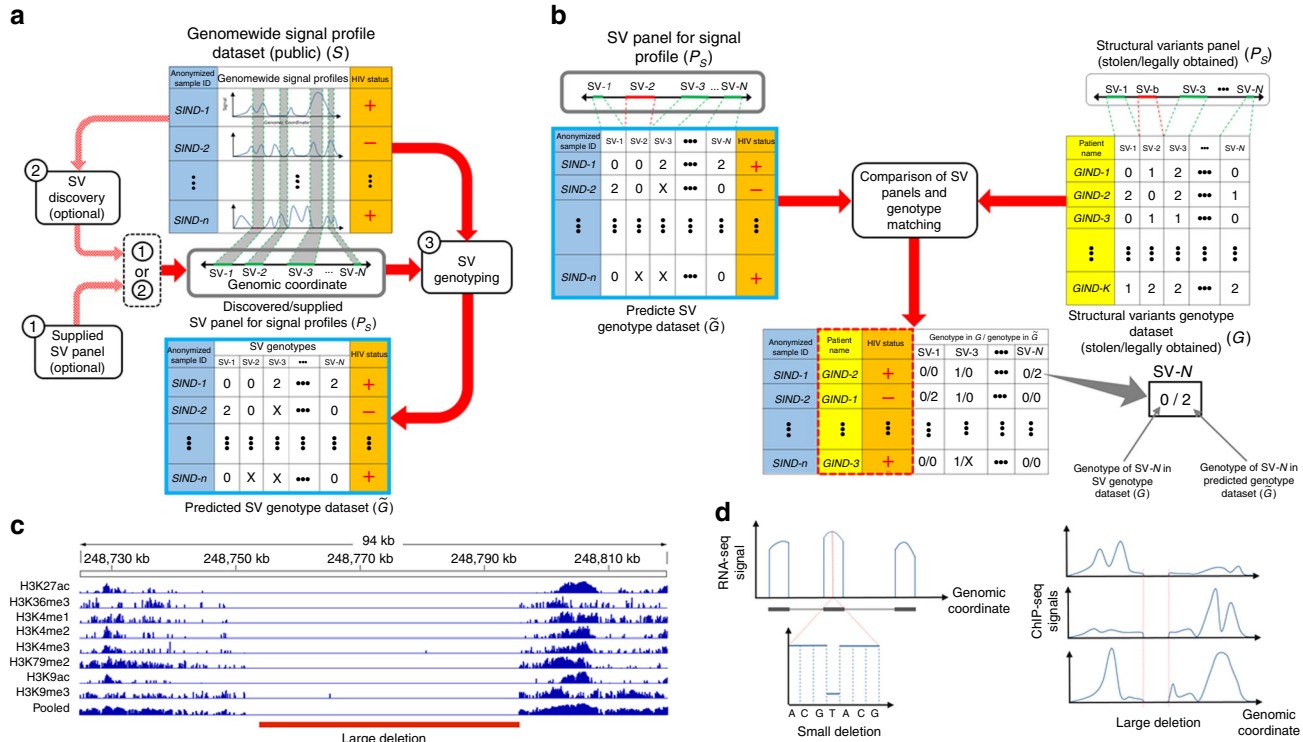

**Fig. 1** Illustration of the attack scenario. **a** The adversary starts the attack with a signal profile dataset ($S$). This dataset contains a genome-wide signal profile and also sensitive information (e.g., HIV status) for each individual. The names are anonymized into IDs as shown in the blue shaded column. The adversary uses an SV panel ($P_S$) in the attack. This panel can be obtained from outside (1) or the adversary can use the genome-wide signal profiles to discover the panel (2), as denoted by the shaded red arrows. The adversary then genotypes the SVs (3) in the panel and builds the dataset for genotyped SVs ($\tilde{G}$). **b** The adversary acquires an SV panel ($P_G$) and genotype dataset ($G$), which contains the genotypes of SVs in the panel for a large number of individuals. In order to link the genotyped SV dataset ($\tilde{G}$) to the SV genotype dataset, the adversary compares their SV panel ($P_S$) to the SV panel ($P_G$). For the matching SVs, the adversary compares the genotypes. The individuals in $G$ who have good matches with respect to genotype distance are linked to signal profile individuals, as indicated by the matching of colored columns. This linking reveals the HIV status of the individuals in the genotype dataset. **c** This example shows a large deletion in the NA12878 individual and how it affects signal profiles. A 70 kb long region is deleted in the NA12878 individual and the decrease in signal profiles show the loss of signal along the deletion. **d** This schematic shows large and small deletions and how they are manifested in signal profiles. The large deletions show a large decrease in the signal profiles, while the small deletions have much smaller footprints

panel of genomic SV loci, which are denoted by $P_S$. For each individual, the adversary utilizes the signal profile and genotypes the deletions in $P_S$. After genotyping, the adversary builds a data matrix with the predicted genotypes, which is denoted by $\tilde{G}$. We refer to this scenario as linking based on "genotyping only". The second scenario, also illustrated in Fig. 1a, is very similar except that the adversary does not have access to, but discovers the panel of SV from the signal profiles. The adversary then uses the signal profiles to genotype the SVs in this de novo-discovered SV panel. We refer to this scenario as linking based on "joint discovery and genotyping". After genotyping, the genotyped SV matrix ($\tilde{G}$) includes the predicted SV genotypes and the sensitive information (e.g., HIV status). $\tilde{G}$ can also be thought of as a noisy genotype matrix, since the genotype predictions may contain errors.

The second step of the linking attack is cross-referencing the individuals in $\tilde{G}$ and the individuals in the $G$ (Fig. 1b). The adversary first compares the genotyped SV panel ($P_S$) to the SV panel of the genotype dataset, which is denoted by $P_S$. After matching the SVs in the two panels, the adversary compares the genotypes of the matching SVs in the two panels. The adversary uses this comparison to cross-reference the individuals in two datasets and finds the individuals that best match each other with respect to genotype match distance. The results are used to link the individuals in genotype dataset to those in the signal profile dataset and the sensitive information, e.g., HIV status of

individuals in the genotype dataset are revealed to the adversary (the matched columns in the final linked matrix).

**Information content and predictability of SV genotypes**. In order to assess the correct predictability of SV genotypes, we propose using genome-wide predictability of SV genotypes, $\pi_{GW}$, from signal profiles. The predictability is defined as the conditional probability of the variant genotype given the signal profile. Predictability measures how accurately an SV genotype can be inferred using the signal profile (Methods section, Supplementary Note 3). By this definition, predictability only depends on the genomic signal levels of an individual and how well they can be used to predict genotypes. For example, Fig. 1c illustrates a large deletion that can be easily predictable using histone modification signal profiles. An important property of $\pi_{GW}$ is that it is computed for each individual independently. Therefore $\pi_{GW}$ is independent of the population frequency of the variant.

To quantify the information content of each SV, we utilize a previously proposed metric termed individual characterizing information (ICI)[15]. For a given variant genotype, ICI measures how much information it supplies for pinpointing an individual in a population. This measure gives higher weight to genotypes that have low population frequency. As the genome-wide predictability is independent of the population frequency of the variants, the adversary can utilize genome-wide prediction

approaches and predict rare variant genotypes to gain high ICI and characterize individuals accurately.

**Linking attacks using RNA-Seq signal profiles**. As we discussed previously, RNA-Seq signal profiles generally have high-depth but low-breadth coverage and they can be used to detect small deletions (<10 bp). We first focused on the predictability of small deletions using RNA-Seq signal profiles. Figure 1d illustrates a hypothetical example of how small deletions in RNA-Seq signal profiles can be detected as small and sudden dips in the signal. As an example showing the effect of small deletions on RNA-Seq signal profiles, we include a screenshot of signal profiles around a small deletion for six individuals in the GTEx Project (Supplementary Fig. 3, Supplementary Note 4). The two base pair deletion, rs34043625, can be easily detected for three of the individuals shown. An important aspect of the effect of small deletions on the signal profile is that they affect the gene's signal profile locally and effects its total expression to a much smaller extent compared to the eQTLs that cause a much larger change (Supplementary Note 5).

Each small deletion is manifested as an abrupt dip in the signal profile (Fig. 1d). The discovery and genotyping of a deletion relies on detecting these dips. We first estimated the genome-wide predictability for the panel of small deletions in 1000 Genomes Project using the RNA-Seq expression signal profiles from the GEUVADIS project. Figure 2a, b shows $\pi_{GW}$ vs. ICI for small deletions. A substantial number of deletions has much higher predictability compared to a dataset where the signal profile is randomized with respect to the location of deletions. In addition, many variants have very high ICI (on the order of 5–6 bits) with high predictability (greater than 80%). This result shows that the signal profile-based attack scenario is much more powerful than other approaches like population-wide prediction of variant genotypes (Supplementary Fig. 4).

In order to present the practicality of using small deletion genotyping, we propose an instantiation of a linking attack where we utilize outlier signal levels in the signal profiles for the discovery and genotyping of small deletions. As mentioned above, the genotyping of deletions is based on detecting abrupt dips in the signal profile. In order to detect these dips, the adversary utilizes a quantity termed self-to-neighbor signal ratio, denoted by $\rho_{[i,j]}$, which measures the extent of the dip in the signal as the fraction of signal on the interval and the signal in the neighborhood,

$$\rho_{[i,j]} = \frac{\text{Average signal within } [i,j]}{\text{Average signal within neighborhood of } [i,j]}.$$

The genomic regions with low $\rho_{[i,j]}$ values point to intervals that tend to have dips in them. For each individual, the prediction method sorts the small deletions with respect to the self-to-neighbor signal ratio and assigns a homozygous genotype to a number of deletions with the smallest self-to-neighbor signal ratio (Methods section). The adversary then compares these genotyped deletions to the genotype dataset and identifies the individual whose deletion genotypes are closest to the predicted genotypes. Using this genotyping strategy, we simulated an attack to link the

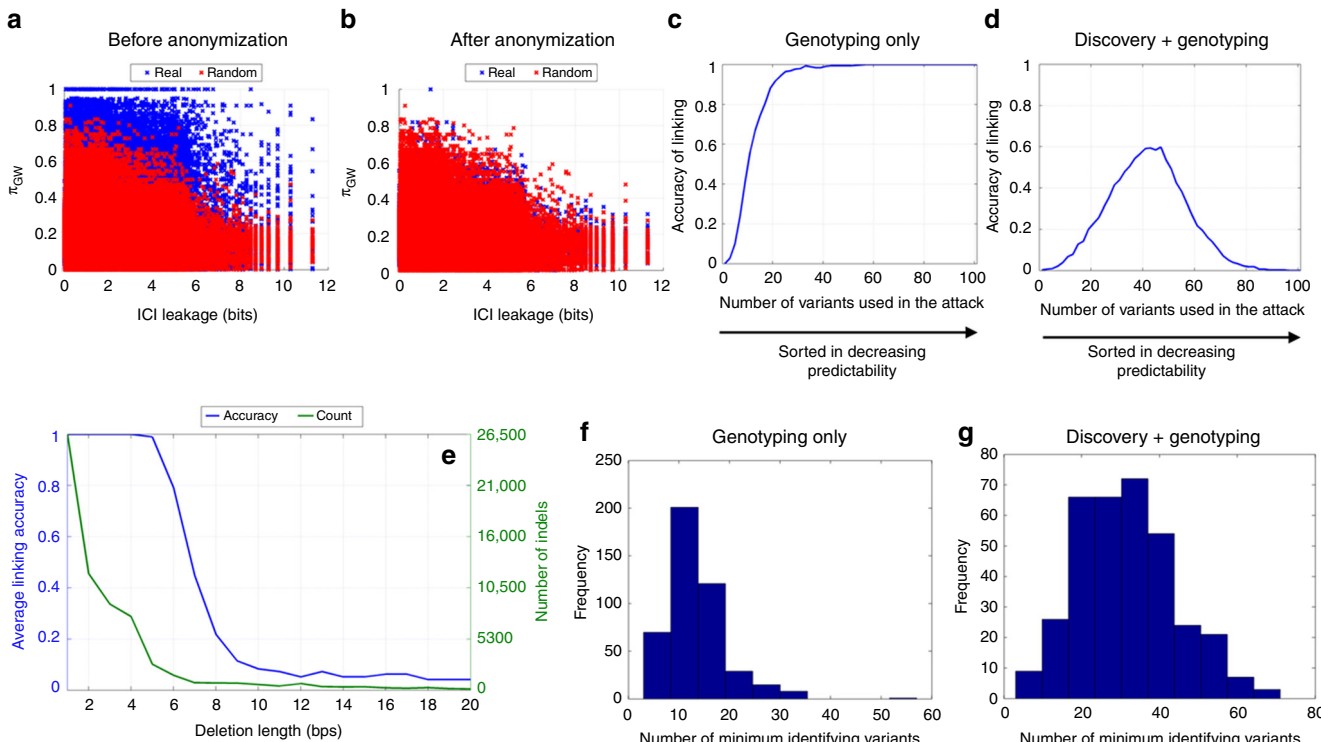

**Fig. 2** The accuracy of linking attack on GEUVADIS dataset. **a** The scatter plot shows the ICI vs. predictability for each deletion (dot). The real data (blue dots) show a much higher predictability compared to randomized data (red dots). **b** After anonymization of signal profiles, the predictability of real data is decreased substantially. **c** This plot demonstrates the accuracy of linking with genotyping of a known panel. The number of variants used in the attack is shown on the x-axis, while accuracy is shown on the y-axis. The variants are sorted with respect to decreasing predictability. **d** This shows the linking accuracy when the adversary performs joint discovery and genotyping of deletions to achieve linking. **e** The blue plot shows the accuracy of linking when indels of a specific length are used in the attack. The green plot shows the distribution of indel lengths. **f** For the genotyping only scenario, the plot shows the distribution of the minimum number of variants required to identify each individual. The x-axis shows the number of indels and the y-axis shows the frequency of individuals that can be identified. **g** For the scenario where adversary discovers the SV panel first and performs genotyping on the discovered panel, the plot shows the distribution of the minimum number of variants required to identify each individual

RNA-seq signal profile dataset from the GEUVADIS Project[28] to the 1000 Genomes Project genotype dataset. In the genotyping only scenario, the linking is perfectly accurate when more than 40 deletions are used (Fig. 2c). In the joint discovery and genotyping scenario, the linking accuracy is maximized (around 60%) when the attacker utilizes the top 50 deletion candidates in linking (Fig. 2d). Next, we studied how accurate the linking would be if the adversary used deletions of different lengths. Figure 2e shows the accuracy and number of insertions and deletions (indels) with different lengths. The accuracy of linking decreases substantially for indels that are longer than 5 bps. The decrease in accuracy is affected by both the decrease in the number of indels (i.e., low ICI), shown in Fig. 2e, and the decreasing predictability of the longer indels. We also found that as small as 30 indels are sufficient to correctly link a large fraction of individuals for genotyping only (Fig. 2f) and for joint discovery and genotyping (Fig. 2g) scenarios.

In the previous analysis, the sample set used for discovery of the deletion panel and the RNA-Seq sample set were matching (i.e., 1000 Genomes individuals). This may cause a bias in linking because the SV genotype dataset may contain rare deletions that are also in the panel of deletions. This would make it trivial to link some of the individuals. To get around this bias, we next studied the linking attacks where the signal profile dataset was generated by the GTEx Project[19,20] and the panel of small deletions was generated by the 1000 Genomes Project, thus utilizing a non-matching set of individuals to decrease biases potentially introduced by the indels panel. In other words, the deletion panel is discovered in a sample set that is totally independent of the sample set that the adversary is linking. With this setup, as before, many deletions were highly predictable (>80%) and very high in ICI (>5 bits) (Fig. 3a, b).

We instantiated the linking attack using an extremity-based approach. In genotyping only scenario, the linking accuracy was close to 100% using a relatively small number of variants (30 variants) (Fig. 3c). Interestingly, as the attacker increases the number of variants used in the attack beyond 30 variants, the linking accuracy decreases. This was caused by the fact that the additional variants were incorrectly genotyped. The additional variants act as noise and cause the linking accuracy to drop.

We also asked whether the adversary can assign reliability scores to the linked individuals. We tested whether the first distance gap (Methods section) measure is suitable for evaluating the reliability of linking. This is important because when the overall linking accuracy is low (e.g., smaller than 50%), unless the attacker has a systematic way of selecting correct linkings, the risk of a linking attack is low. As a test case, we focused on a linking where the adversary used 200 deletions with an overall linking accuracy of 35% (Fig. 3c). Figure 3d shows the sensitivity and positive predictive value (PPV) with a changing first distance gap metric. The adversary could link 10% of the individuals perfectly, and 20% with around 90% accuracy. Figure 3d also shows the average sensitivity and average PPV over 100 random selections of the linkings. As expected, the PPV was always around 35% and average sensitivity was always smaller than for a first distance gap-based selection of linkings. These results show that even though some parameter selections (e.g., number of variants) may show low accuracy, the adversary can increase accuracy by selecting the linkings using the first distance gap measure.

**Linking attacks using ChIP-Seq signal profiles**. We next focused on predictability vs. ICI of deletions over 1000 base pairs. Since the deletions are large, the signal profiles that are suitable for genotyping them must have high-breadth coverage. We utilized

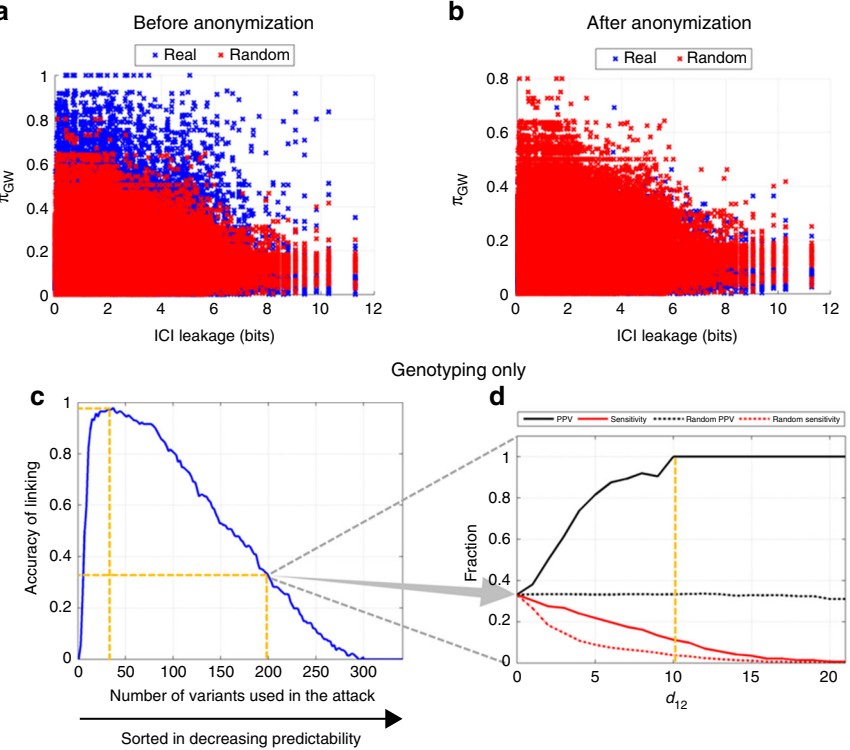

**Fig. 3** The accuracy of linking attack on GTEx dataset. **a**, **b** The ICI leakage vs. predictability for all the indels before (**a**) and after (**b**) signal profile anonymization. **c** The linking attack accuracy with a changing number of variants used in the attack. The x-axis shows the number of variants used in the attack and the y-axis shows the accuracy of linking. **d** When the adversary uses 200 variants in **c** and selects linking based on thresholding $d_{1,2}$ (shown on x-axis), the plot shows on the y-axis the sensitivity (black) and positive predictive value (red) of linkings for real (solid) and random (dashed) datasets while $d_{1,2}$ is changed

ChIP-Seq signal profiles for histone modifications, which generally manifest high-breadth and low-depth coverage. Several recent studies have generated individual-level epigenomic signal profiles through ChIP-Seq experiments[29–31] (Supplementary Note 4, 5). These studies aimed at revealing how genetic variation interacts with epigenomic signals, mainly histone modifications. These datasets are very convenient for our study because the majority of the individuals have matching SV genotype information in the 1000 Genomes Project. Although we are focusing on the predictability of large deletion genotypes from ChIP-Seq profiles, they can also be used for small deletion genotyping where there is large enough depth of coverage.

We used the personal epigenomic signal profiles (Kasowski et al.[31] and Kilpinen et al.[30]) to quantify how much characterizing information leakage they provide. Whenever multiple histone modifications are available for an individual, they are pooled. We computed $\pi_{GW}$ vs. ICI using the panel of large deletions in 1000 Genomes Project. Figure 4a, b shows $\pi_{GW}$ vs. ICI for the large deletions in the 1000 Genomes Project. (Methods section). Similar to our small deletion analysis, for both datasets there were many large deletions with high predictability and high ICI.

We next focused on instantiating linking attacks using ChIP-Seq profiles. We again utilized a variant of the outlier-based genotyping in the linking attack. The genotyping of the panel of large deletions was performed as follows. The average ChIP-Seq signal on each deletion was computed and the variants were sorted with respect to their average signal in increasing order. The deletions with smallest ChIP-Seq signal were assigned a homozygous deletion genotype (Methods section). For the deletions with assigned genotypes, we identified the individual in the genotype dataset (from the 1000 Genomes project) whose genotypes match closest to the assigned genotypes. Figure 4c shows the accuracy of a linking attack based on the genotyping only scenario, where the adversary was assumed to have access to the panel of variants from the 1000 Genomes Project. The linking accuracy reached 100% with a fairly small number of deletions for both datasets. For the joint discovery and genotyping scenario, where the adversary first discovered deletions and then genotyped them, the accuracy was also very high with a small number of identified deletions (Fig. 4d).

We then aimed to evaluate the leakage from combinations of histones. We focused on the individual NA12878, for which there is an extensive set of histone modification ChIP-Seq data from the ENCODE project[17] (Supplementary Note 4). We evaluated whether different combinations of histone modifications render NA12878 vulnerable against a linking attack among 1000 Genomes individuals, as illustrated in Fig. 4e. Results show that NA12878 is vulnerable when attack uses the combinations that cover the largest space in the genome. This can be simply explained by the fact that, when a histone marks cover a larger

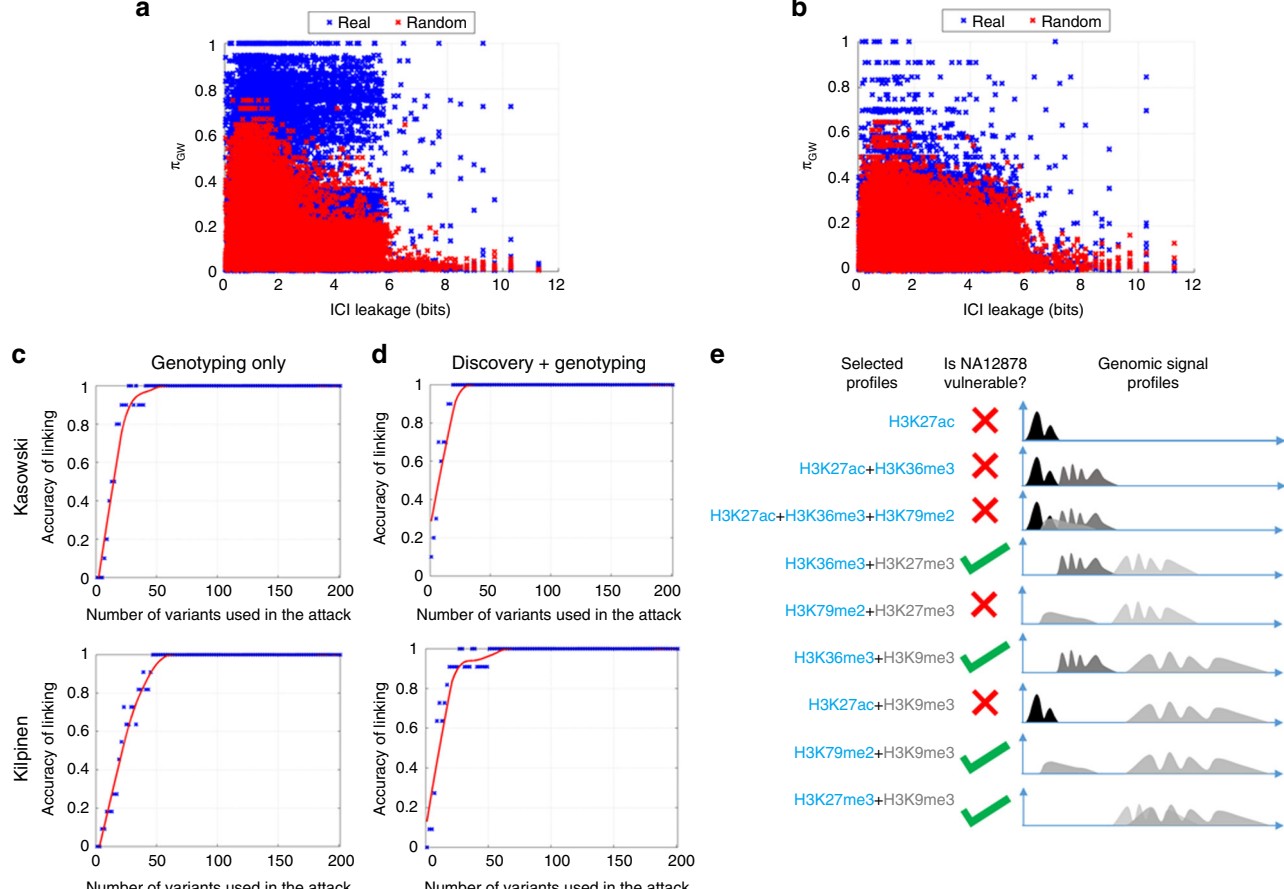

**Fig. 4** Leakage from ChIP-Seq signal profiles through small deletions. **a**, **b** Scatter plots show ICI leakage vs. predictability for Kasowski (**a**) and Kilpinen (**b**) datasets. **c** The accuracy of linking attack on the two datasets for a genotyping only scenario. The x-axis shows the changing number of variants used in the attack and the y-axis shows the linking accuracy. **d** The accuracy of linking on the two datasets when the adversary performs the attack by joint discovery and genotyping of deletions. **e** The accuracy of linking of NA12878 when adversary utilizes different combinations of histone modifications. The first column shows different combinations. The middle column indicates whether NA12878 is identifiable among 1000 Genomes samples, represented by green check for yes and red cross for no. The third column is a schematic representation of signal profiles for each combination

genomic region, a larger number of deletions can be detected and genotyped. For example, H3K36me3 and H3K27me3, an activating and a repressive mark, respectively, are mainly complementary to each other and render NA12878 vulnerable. In addition, H3K9me3, a repressive mark that expands very broad genomic regions, renders NA12878 vulnerable in several combinations with other marks. By contrast, H3K27ac, an activating histone mark that spans punctate regions, does not render NA12878 vulnerable.

**Linking attacks using Hi-C matrices**. We next tested whether a relatively new data type, Hi-C interaction matrices, can be used for the identification of genomic deletions. Hi-C is a high-throughput method for identifying long-range genomic interactions and three-dimensional chromatin structure[32]. After raw Hi-C data is processed, it is converted to a matrix where the entry ($i$, $j$) represents the strength of interaction between $i$th and $j$th genomic positions. To study leakage from Hi-C matrices, we again focused on the NA12878 individual for whom Hi-C interaction matrices were generated at different resolutions[33]. We first converted the matrix into a genomic signal profile by summing the matrix along columns (Fig. 5a, Methods section). It is worth noting that the standard analysis of Hi-C matrices does not involve such a signal profile generation. Using the signal profile, we simulated an extremity-based linking attack using the outliers in the Hi-C signal profile. Similar to ChIP-Seq profile-based attack, we genotyped the 1000 variants from the 1000 Genomes Project Panel (Methods section). We next compared the predicted genotypes with all the genotypes in the 1000 Genomes Project. We observed that NA12878 was vulnerable to attack when the Hi-C contact matrix resolution was 10 kb or smaller (Fig. 5b).

It is important to clarify that we are focusing on using the final output of Hi-C data, that is the Hi-C contact matrix, for generating a genome-wide signal profile and performing a linking attack. We are not studying the possibility of discovering complex SVs using the paired-end reads of a Hi-C experiment, which is a separate issue[34]. This would require access to mapped reads, which we assume the attacker does not have. As we explained above, our attack scenario treats the Hi-C data as any type of sequencing data and uses the linear genomic signal profile to identify deletions for the purpose of linking datasets. The results highlight the fact that Hi-C interaction matrices themselves may leak substantial amounts of characterizing information.

**Anonymization of RNA-Seq signal profiles**. An important aspect of genomic privacy is risk management and the protection of datasets. Anonymization of the datasets is the most effective way to ensure safe protection when sharing data publicly (Supplementary Note 6). Personal RNA-Seq datasets are currently by far the most abundant functional genomic datasets. For example, RNA-Seq signal profiles are being publicly shared from the GTEx project, although the genotypes are not in public access. In addition, RNA-Seq is becoming commonly used in the clinical settings and new RNA-Seq-based assays are being developed to probe gene expression, for example single-cell RNA-Seq. Altogether, these factors make the protection of RNA-Seq data urgent. The most effective way to protect against a linking attack scenario is to ensure that deletion genotypes cannot be inferred from signal profiles. As we showed in the previous sections, small deletions are a major source of leakage of genetic information from RNA-Seq signal profiles. We propose systematically removing the dips in signal profiles as a way to anonymize the profiles against the prediction of small deletions (Methods section). We observed that this procedure removes the dips in the signal effectively while conserving the signal structure fairly well. To evaluate the effectiveness of this method, we applied signal profile anonymization to the RNA-Seq signal profiles generated by the GEUVADIS and GTEx Project consortia. After applying signal profile anonymization, the large fraction of the leakage was removed for the GTEx datasets (Figs. 2b and 3b). We also observed that the extremity-based linking attack proposed in the previous section was ineffective in characterizing individuals such that no individuals were vulnerable for the GTEx project and only 1% of the individuals were vulnerable for the GEUVADIS dataset. Importantly, this procedure can be used for anonymizing not only RNA-Seq signal profiles but also other signal profiles against attacks based on small deletion genotyping. However, the anonymization is not as effective for large deletions. This is not a major concern for RNA-Seq signal profiles, as we observed that large deletions were not easily genotyped using RNA-Seq data. However, as we showed in the previous section, linking attacks can be successful when they use large deletions that are genotyped using ChIP-Seq datasets.

## Discussion

Sequencing-based functional genomics assays provide a large amount of biological information for understanding the dynamic nature of gene activity and epigenetic regulation. This information is extremely valuable for understanding genetic mechanisms behind disease initiation and progression. Thus, data producers and owners want to share these data as openly as possible (Supplementary Note 6). At the same time, genomic data can contain variant genotype information within the raw reads that may cause concerns for privacy. These two competing factors, the incentive to share and privacy concerns, make it necessary to carefully evaluate the sharing mechanisms of functional genomics data. To decrease genetic variant leakage in sequencing data, aggregate data formats have been widely used. Two examples are signal profiles and gene expression quantifications. Unlike raw reads, these data do not immediately reveal variant information and are generally accepted to be safe for public data sharing. However, gene expression levels have been shown to leak enough genotype data to be used in accurate linking attacks[15,21]. In this study, we evaluated the possible privacy concerns around sharing signal profiles.

We systematically analyzed a critical source of sensitive information leakage from signal profile datasets, which were previously thought to be largely secure. Our results show that an adversary can perform very accurate linking attacks for

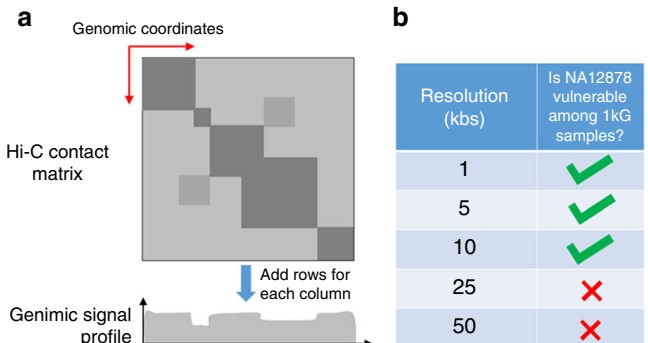

**Fig. 5** Representation of the linking attack that utilizes Hi-C interaction matrix data. **a** Schematic representation of how genome-wide signal profile is computed from the interaction matrix. Each column $i$ of the matrix is summed along the rows and the total value is recorded at the $i$th entry of the signal profile. **b** Table shows whether NA12878 is vulnerable when different resolutions of the interaction matrix is used in linking. A green check indicates that NA12878 is vulnerable while a red cross indicates it is not vulnerable

characterizing individuals by the genotyping of SVs using functional genomics signal profiles. These results reflect how the rich nature of functional genomics data can cause privacy concerns in an unforeseen manner. This is an interesting aspect of the data. Although there may be some variant information in functional genomics signal profiles, these data are not generated mainly for detecting variant information. The main purpose of them is to reveal how they change under different conditions and how they relate to phenotypes, which may be sensitive. The existence of residual variant information, as we showed, may enable an adversary to reveal sensitive information about an individual.

Although we focused mainly on RNA-Seq and ChIP-Seq signal profiles, the linking attack scenario and the measures that we presented are data-driven and are generally applicable to any type of genome-wide signal profile. For example, linking attacks can be easily carried out on DNA sequencing signal profiles. In addition, signal profiles from genome-wide profiling techniques other than sequencing-based assays, like ChIP and expression tiling arrays[35,36], can be vulnerable to the linking attack scenario that we presented. Different genome-wide data representations (e.g., Hi-C interaction matrices) can be utilized for the generation of genome-wide signal profiles; these can in turn leak sensitive information. We believe that many more genome-wide omics technologies will be developed in the near future[37]. Genome-wide signal profiles will be a vital source of information in the analysis of these datasets. The framework we presented here can be utilized for assessing the leakage and protection of these datasets. In addition, albeit our focus is on small and large deletion variants, the analyses of predictability and practical linking attacks can be extended for other SVs such as genomic insertions.

We showed that linking can be done by predicting a fairly small number of variants (generally less than 100). Our results show that these data leak enough information for individual characterization among a large set of individuals. This can cause practical privacy issues because several large consortia are making signal profiles publicly available. For example, GTEx RNA-Seq signal profiles are publicly available through the University of California, Santa Cruz (UCSC) Genome Browser. Given the extent of public sharing of datasets, we believe that the anonymization of RNA-Seq signal profiles using the signal processing technique that we proposed is very useful. Our method applies signal smoothing around all the known deletions and removes a significant amount of characterizing information. The anonymization procedure can be easily integrated into existing functional genomics data analysis pipelines. We believe that this anonymization technique can complement other approaches for removing genetic information from shared datasets. For example, file formats like MRF[38] and tagAlign[17] can enable removing raw sequence information from reads while keeping the information about read mapping intact. While the anonymization method is effective for closing the leakage from signal profile, it does not close all sources of leakages. Additional sources of leakage can be present in the raw reads and gene expression levels as demonstrated in earlier studies[15,21]. In addition, complex machine learning techniques can be used to predicted genotypes using higher level data such as pathway activity[20,39].

## Methods

We provide the details of the computational methodologies[39]. We first introduce the notations. The genomic deletions are intervals of genomic coordinates. We refer to them simply as intervals, for example, a deletion between genomic positions $i$ and $j$ by $[i, j]$. The genotype of a genomic deletion at $[i, j]$ is denoted by $G_{[i,j]}$, which is a discrete random variable distributed over the three values $\{0, 1, 2\}$. These values correspond to the three genotypes of the deletion and represent how many copies of the genomic sequence are deleted. The functional genomics read depth signal is denoted by $\mathbf{S}$, which is a vector of values corresponding to each genomic position. The signal level at genomic position $i$ is denoted by $S_i$. An important

quantity that we utilize in formulating methods is the multi-mappability profile of the deletion regions. Multi-mappability is a signal profile that measures, for each position in the genome, how uniquely we can map reads. The multi-mappability signal is denoted by $\mathbf{M}$, which is a vector of multi-mappability signals for all the genomic positions, and the signal at genomic position $i$ is denoted by $M_i$. The multi-mappability signal profile is generated as follows: the genome is cut into fragments and the fragments are mapped back to the genome using bowtie2[40] allowing the multi-mapping reads. We then generate the read depth signal of the mapped reads. In this signal profile, the uniquely mapping regions receive low signal while the multi-mapping regions receive high signal[41].

**Predictability of genotypes and characterizing information**. The genome-wide predictability, $\pi_{\mathrm{GW}}$, of a deletion genotype refers to how well a deletion can be genotyped given the functional genomics signal ($\mathbf{S}$) of interest. We assume that the adversary employs a prediction methodology based on statistical modeling of the deletion genotypes with respect to read depth signal profile such that the adversary utilizes features from the functional genomics signal profile. We define here the features that are most useful for genotyping deletions (Supplementary Fig. 5). Given a deletion $[i, j]$, an important feature for genotyping the deletion is the average functional genomic signal within the deletion:

$$\bar{s}_{[i,j]} = \frac{\sum_{i'=i}^{j} S_{i'}}{j - i + 1}.$$

Another feature is the average multi-mappability signal within the deletion:

$$\bar{m}_{[i,j]} = \frac{\sum_{i'=i}^{j} M_{i'}}{j - i + 1}.$$

In order to measure the extent of the dip within the signal, we observed that a measure we termed self-to-neighbor signal ratio and neighbor signal balance ratio are very useful for genotyping. Given a deletion $[i, j]$, self-to-neighbor signal ratio, denoted by $\rho_{[i,j]}$, is computed as

$$\rho_{[i,j]} = \frac{2 \times \bar{s}_{[i,j]}}{\bar{s}_{[2i-j+1,i-1]} + \bar{s}_{[j+1,2j-i+1]}}.$$

This is simply twice the ratio of total signal on the deletion and the total signal in the neighborhood of the deletion. The neighbor signal balance ratio is computed as

$$\eta_{[i,j]} = \min\left(\frac{\bar{s}_{[j+1,2j-i+1]}}{\bar{s}_{[2i-j+1,i-1]}}, \frac{\bar{s}_{[2i-j+1,i-1]}}{\bar{s}_{[j+1,2j-i+1]}}\right).$$

Finally, we observed that the average signal on the neighborhood of the deletion coordinates are useful in genotyping deletions. This is because when the neighbor signals are more balanced around a dip, that is, higher $\eta_{[i,j]}$, the accuracy of deletion genotyping is higher. Next, we computed the average signal in the neighborhood as

$$\tau_{[i,j]} = 0.5 \times \left(\bar{s}_{[2i-j+1,i-1]} + \bar{s}_{[j+1,2j-i+1]}\right).$$

We defined $\pi_{\mathrm{GW}}$ as the conditional probability of a deletion genotype $g$ given the five features computed from a functional genomics signal profile:

$$\pi_{\mathrm{GW}}\left(G_{[i,j]} = g, \mathbf{S}_{[i,j]}\right) = P_{\mathrm{GW}}\left(G_{[i,j]} = g \left| \begin{array}{l} \log_2\left(\bar{s}_{[i,j]}\right), \\ \log_2\left(\bar{m}_{[i,j]}\right), \\ \log_2\left(\rho_{[i,j]}\right), \\ \log_2\left(\eta_{[i,j]}\right), \\ \log_2\left(\tau_{[i,j]}\right) \end{array} \right.\right).$$

This corresponds to the conditional probability (over all deletions within the genome) that we observed for genotype $g$ for a deletion at $[i, j]$ given the average functional genomics signal and average multi-mappability signal over the interval $[i, j]$. The probability is defined over the genome; that is, we estimate the probability for all the deletions in the genome. For this, we computed five features for every deletion in the genome, and then estimated the conditional probability using this set as the sample of deletions.

The basic idea behind the formulation of predictability is the observation that the regions with low functional genomics signal, low multi-mappability (i.e., uniquely mappable), low self-to-neighbor signal ratio, and high average neighbor signal are more likely to be deleted (i.e., their probability is large). Therefore, $\pi_{\mathrm{GW}}$ is higher for deletions that are easier to identify than the deletions with lower $\pi_{\mathrm{GW}}$. In order to estimate the conditional probabilities, we binned the feature values by computing the logarithm and then rounding this value to the closest smaller integer value.

**Discovery and genotyping of deletions from signal profiles.** The practical instantiation of the linking attacks that we studied are based on genotyping the panel of small deletions, $P_S$, using functional genomics data. In addition, when the deletions panel $P_S$ is not available, the adversary also discovers the deletions using the signal profile. For GEUVADIS and GTEx datasets, we performed small deletion genotyping using RNA-Seq signal profiles. The basic idea behind genotyping of deletions is the fact that there is a sudden dip in signal profile whenever there is a deletion (Fig. 1d). In order to detect these dips, we observed that the self-to-neighbor signal ratio is very useful for genotyping small deletions. For all small deletions, we computed self-to-neighbor signal ratio, $\rho_{[i,j]}$, neighbor signal balance, $\eta_{[i,j]}$, and average neighbor signal. We then selected the deletions that satisfied the following criteria:

$$\bar{m}_{[i,j]} < \bar{m}_{\max} (\text{HighMappability})$$

$$\tau_{[i,j]} > \tau_{\min} (\text{High Neighbor Signal})$$

$$\eta_{[i,j]} > \eta_{\min} (\text{High Neighbor Signal Balance})$$

We sorted the set of small deletions that passed these criteria with respect to increasing $\rho_{[i,j]}$. The deletions that are at the top of the sorted list correspond to deletions that are highly mappable (low multi-mappability signal), have strong neighbor signal support (high average neighbor signal), and have a strong signal dip on them (low $\rho_{[i,j]}$, and high $\eta_{[i,j]}$). We selected the top $n$ deletions and assigned them homozygous genotypes, i.e., $G_{[i,j]} = 0$. The basic idea is that the deletions with strongest signal dips are enriched in homozygous deletions. It is worth noting that this genotyping method only assigns homozygous genotypes. Although this might result in low genotyping accuracy (Supplementary Fig. 6), these genotyping predictions have enough information for accurate linking attacks.

We utilize pooled ChIP-Seq read depth signal profiles and Hi-C signal profiles for genotyping large deletions. For genotyping large deletions, we first computed the average signal $\left(\bar{s}_{[i,j]} = \frac{\sum_{i'=i}^{j} \mathbf{S}_{i'}}{j-i+1}\right)$ and average multi-mappability signal $\left(\bar{m}_{[i,j]} = \frac{\sum_{i'=i}^{j} \mathbf{M}_{i'}}{j-i+1}\right)$ on each large deletion. We selected candidate large deletions using average multi-mappability signal:

$$\bar{m}_{[i,j]} < \bar{m}_{\max} (\text{High Mappability})$$

We sorted the deletions that satisfied the above criteria with respect to increasing average signal, $\bar{s}_{[i,j]}$. For the top $n$ deletions, we assigned homozygous genotypes, i.e., $\tilde{G}_{[i,j]} = 0$.

We generally observed that the parameter selection for filtering variants did not have a substantial effect on accuracy of linking attacks as long as they were not made too stringent. In the computational experiments, we used $\bar{m}_{\max} = 1.5$, $\tau_{\min} = 10$, $\eta_{\min} = 0.5$ as the parameter set.

For the case when the adversary does not have access to the deletion panel, we fragmented the genome into windows and used these windows as candidate deletions. We utilized the above procedure for selection of the candidate deletions, which were assigned homozygous deletion genotypes. For small deletions, we used five base pair windows within the exonic regions. For large deletions, we used 1000 base pair windows over the whole genome.

**Instantiations of genome-wide linking attack.** Following genotyping of the deletions in $P_S$, we used the genotyped deletions to link the individual to the individuals in the SV genotype dataset. Given the genotyped deletions for the $k$th individual in the signal profile dataset, we first compared these deletions to the panel of deletions in the genotype dataset, $P_G$. The comparison was performed by overlapping the deletions in $P_S$ and $P_G$. Any two deletions that overlapped at least one base pair were assumed to be common in the two panels. For the "common" set, $\{[i_1, j_1], [i_2, j_2], \dots, [i_n, j_n]\}$, we computed the genotype distance by matching the genotypes,

$$d_{k-l} = \sum_{\substack{a = [i', j'] \in \\ \{[i_1, j_1], \\ \dots \\ [i_n, j_n]\}}} d\left(\tilde{G}_{[i',j']}^{(k)}, G_{[i',j']}^{(l)}\right)$$

where $d_{k-l}$ represents the genotype distance of $k$th individual in the signal profile dataset to the $l$th individual in the genotype dataset and $d\left(G_{[i',j']}, G_{[i',j']}\right)$ is the distance function:

$$d\left(\tilde{G}_{[i',j']}^{(k)}, G_{[i',j']}^{(l)}\right) = \begin{cases} 1 \text{ if } \tilde{G}_{[i',j']}^{(k)} \neq G_{[i',j']}^{(l)} \\ 0 \text{ if } \tilde{G}_{[i',j']}^{(k)} = G_{[i',j']}^{(l)} \end{cases}.$$

We next computed the genotype distance of the $k$th individual to all the individuals in the genotype dataset; $d_{k-l}$ for all $l$ in $[1,k]$ where $k$ represents the number of individuals in the genotype dataset. The individual in the genotype dataset that has the smallest genotype distance was linked to $k$th individual:

$$\text{linked individual's index} = \operatorname*{argmin}_{l' \in [1,K]}(d_{k-l'})$$

Finally, if the linked individual in the genotype dataset matched the individual in the signal profile dataset, we marked the individual in the signal profile as a vulnerable individual. We also computed the first distance gap, $d_{1,2}$, for each linked individual[15] to evaluate the reliability of linking. For a linked individual, the first distance gap is computed as

$$d_{1,2} = d_k^{(1)} - d_k^{(2)}$$

where $d_k^{(1)}$ and $d_k^{(2)}$ are the minimum and second minimum genotype distance among all the genotype distances computed between the $K$th individual and all the genotype dataset individuals.

**Computation of sensitivity and positive predictive value.** In order to compute the sensitivity and PPV of linkings when the linkings are selected using the first distance gap measure, we used following formula:

$$\text{Sensitivity} = \frac{\text{Number of correctly linked individuals with } d_{1,2} > d_{1,2}^{\min}}{\text{Number of All Individuals}}$$

$$\text{PPV} = \frac{\text{Number of correctly linked individuals with } d_{1,2} > d_{1,2}^{\min}}{\text{Number of Individuals with } d_{1,2} > d_{1,2}^{\min}}$$

where $d_{1,2}^{\min}$ represents the minimum first distance gap measure that was used to select individuals. In these formulae, sensitivity represents the fraction of all individuals that the adversary correctly links. PPV represents the fraction of individuals that are correctly linked among the individuals whose linking satisfies the minimum first distance gap threshold.

**Anonymization of signal profile datasets.** The anonymization of the signal profile datasets refers to the process of protecting the signal profile data against correct predictability of the genotypes for deletion variants. As we discussed earlier, the large and small dips in the functional genomics signal profiles are the main predictors of deletion variant genotypes. To remove these dips systematically, we propose using median filtering[42] based signal processing to locally smooth the signal profile around the deletion. This signal processing technique has been used to remove shot noise in two-dimensional imaging data and one-dimensional audio signals[41,43]. For each genomic $a$ in the deletion $[i, j]$, we replaced the signal level using the median filtered signal level:

$$\tilde{x}_a = \text{median}\left(\{x_b\}, b \in \left[a - \frac{l}{2}, a + \frac{l}{2}\right]\right)$$

where $x_a$ refers to the signal level at the genomic position $a$, $l = j - 1 + 1$, $\tilde{x}_a$ refers to the smoothed signal level at position $a$, and median refers to the median of all the signal values in the genomic region $\left[a - \frac{l}{2}, a + \frac{l}{2}\right]$. The median is computed by sorting all the signal levels and choosing the value in the middle of the sorted list of signal levels.

**Data availability.** The mapped reads for the RNA-Seq data from the GEUVADIS project were obtained from the GEUVADIS project website (http://geuvadis.org/). We filtered out the individuals for which there are 1000 Genomes genotypes available. This filtering yielded 421 individuals. The RNA-Seq mapped reads from the GTEx project were obtained from the dbGAP portal using the dbGAP accession number phs000424. We used only the RNA-Seq datasets from whole blood tissue to create signal profiles. These datasets are used in analysis of the linking attacks that used RNA-seq signal profile datasets.

The SV panel and genotypes were obtained from the 1000 Genomes Project (http://www.internationalgenome.org/data). The genotypes for 2504 individuals are used. Very low frequency SVs may introduce bias since they can uniquely identify an individual. In order to get around this bias, we removed the SVs of which the minimum genotype frequency was larger than 0.01. In addition, we extended the genotype dataset by re-sampling the 1000 Genomes deletion dataset and created genotype data for 10,000 simulated individuals. For the analysis that used dbGAP datasets, we obtained small deletion genotypes from dbGAP (accession number phs000424). These datasets are used in analysis of the linking attacks that used RNA-seq signal profile datasets.

The histone modification datasets from Kilpinen et al. are obtained from ArrayExpress website with accession number E-MTAB-1884. Kasowski et al. datasets are obtained by contacting the authors of the study. In total, we obtained 27 individuals from these datasets. These are used in analysis of linking attacks using ChIP-Seq signal profile datasets. The genotype dataset is obtained from the 1000 Genomes project as explained above.

We utilized randomized datasets to compare predictability with real data. In order to create randomized data, we shuffled the signal profiles circularly. In this way, the association between the SV genotypes and signal profiles were randomized.

The source code for linking attacks and anonymization can be obtained from http://privasig.gersteinlab.org. All the above data can also be obtained by contacting the authors.

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

## Acknowledgments

We thank Akdes Serin Harmanci for critical comments on the method development and analysis of the data.

## Author contributions

A.H. designed and performed the experiments. A.H. drafted the manuscript. A.H. and M. G. wrote the manuscript. Both authors read and approved the final manuscript.

## Additional information

**Competing interests:** The authors declare no competing interests.

