## [Peer Review File · Nature Communications]

REVIEWERS' COMMENTS:

Reviewer #1 (Remarks to the Author):

I was one of the original reviewers for this manuscript when it was under consideration at Nature Methods. I found the work interesting and sound, albeit with room for improvement and clarification. I have carefully read the the authors' response and revised manuscript, I am happy with the revision and do not have additional comments.

Reviewer #2 (Remarks to the Author):

The authors have largely addressed prior concerns

RESPONSE TO REVIEWERS' COMMENTS FOR "ANALYSIS OF SENSITIVE INFORMATION LEAKAGE IN FUNCTIONAL GENOMICS SIGNAL PROFILES THROUGH GENOMIC DELETIONS"

RESPONSE LETTER

-- Ref1: Final comments --

Reviewer Comment	I was one of the original reviewers for this manuscript when it was under consideration at Nature Methods. I found the work interesting and sound, albeit with room for improvement and clarification. I have carefully read the the authors' response and revised manuscript, I am happy with the revision and do not have additional comments.
Author Response	We sincerely thank the reviewer and the editor for the constructive comments, which we believe made our paper stronger.

-- Ref2: Final comments --

Reviewer Comment	The authors have largely addressed prior concerns
Author Response	We sincerely thank the reviewer for the constructive criticism.